# Association between History of Prolonged Exclusive Breast-Feeding and the Lung Function Indices in Childhood

**DOI:** 10.3390/children9111708

**Published:** 2022-11-08

**Authors:** Evanthia P. Perikleous, Sotirios Fouzas, Magdalena Michailidou, Anna Patsourou, Dimos Tsalkidis, Paschalis Steiropoulos, Evangelia Nena, Athanasios Chatzimichael, Emmanouil Paraskakis

**Affiliations:** 1Medical School, Democritus University of Thrace, 691 00 Alexandroupolis, Greece; 2Pediatric Respiratory Unit, University Hospital of Patras, 26504 Patras, Greece; 3Department of Pneumonology, Medical School, Democritus University of Thrace, 691 00 Alexandroupolis, Greece; 4Laboratory of Social Medicine, Medical School, Democritus University of Thrace, 691 00 Alexandroupolis, Greece; 5Department of Pediatrics, Medical School, Democritus University of Thrace, 691 00 Alexandroupolis, Greece; 6Pediatric Respiratory Unit, Pediatric Department, University of Crete, 700 13 Heraklion, Greece

**Keywords:** breastfeeding, spirometry, children

## Abstract

Although the propitious effects of breastfeeding on children’s health are indisputable, the impact of exclusive breastfeeding on the lung function later in life remains controversial. Our objective was to explore the possible associations between breastfeeding and the lung function of children who were exclusively breastfed for an extensive period of time. This was a cross-sectional study of children who were exclusively breastfed for more than 12 months. Demographics and anthropometric data were collected; the body mass index (BMI), % body fat, and % central obesity were calculated; and all the participants underwent standard spirometry with reversibility testing. The relationship between breastfeeding duration and spirometric parameters was assessed by Spearman’s correlation and multivariable regression, after adjustment for other confounders. Forty-six children (21 boys), aged 9.2 ± 2.4 years, with a reported breastfeeding duration of 27.5 ± 12.5 months (range 12–60 months) were included; 13% were overweight (none were obese) and 21.7% had central obesity. The average FEV1 was 104.7 ± 10.4% and the average FEF25-75 was 107.9 ± 13.3%. The duration of exclusive breastfeeding was positively correlated with FEF25-75% (r = 0.422, *p* = 0.003). Multivariable linear regression analysis confirmed the above finding (beta coefficient 0.478, *p* = 0.002), independently of age, overweight, and central obesity. No correlation was noted between the duration of breastfeeding and other spirometric parameters. In addition to its favorable impact on the metabolic profile, prolonged exclusive breastfeeding seems to exert a propitious effect on the function of smaller airways throughout childhood.

## 1. Introduction

Ample evidence from epidemiological and biological studies has emphasized the propitious health effects of breastfeeding [1]. However, the prevalence of breastfeeding is often influenced by many factors, such as socio-demographic features, mode of delivery, early time of first breastfeeding, and the presence of personnel trained and skilled to support human lactation, which has been demonstrated to be essential for both the initiation and longer duration of breastfeeding [2,3,4,5]. The American Academy of Pediatrics (AAP), in consistence with guidelines and policies of the World Health Organization (WHO) both recommend that infants be exclusively breastfed for approximately 6 months; and furthermore, support continuing breastfeeding for 2 years or beyond, in order to induce the optimal short- and long-term outcomes for the mother–child dyad [5,6].

Initially, the vast majority of women commence breastfeeding soon after delivery; thus, the great challenge is, as a society and as healthcare professionals, to establish and protect the continuation of this very positive initial step. A striking paradigm, addressing this goal in the best possible way, is the implementation of the ten steps towards successful breastfeeding of the WHO and United Nations Children’s Fund (UNICEF) [7]. Hence, breastfeeding should be considered as an important public health priority, and it is mandatory for pediatricians to engage in advocating and supporting ideal breastfeeding practices.

In Greece, the latest national data showed an increase in breastfeeding rates [8,9], although the COVID-19 pandemic has influenced breastfeeding initiation and duration positively [9]; unfortunately, Greece has a lack of congruence with the WHO recommendation, as the exclusive breastfeeding percentage at 6 months old continues to be unsatisfactory. The key determinant is what is commonly believed regarding baby-feeding habits, such as human milk substitutes and solids administration in early infancy, in the first months of life, and after the 4th month of life, respectively, which is considered the ‘norm’ [8,9]. An admirable effort towards altering the prevailing baby-feeding attitudes is the ongoing national program ‘Alkyoni’, an initiative aiming to protect, promote, and support breastfeeding through numerous awareness actions.

The protective action of human milk against respiratory and gastrointestinal infections in infancy, together with its favorable effects against the development of obesity, diabetes, and atopy in childhood and adolescence, have been well established [1,10,11,12]. Evidence from observational studies also suggests that breastfeeding minimizes the lung sequelae of respiratory infections [13,14,15,16], results in improved lung function at school age [17], especially in children with an atopic background [18] and preserves lung function in those exposed to significant air pollution [19], including passive smoking [20], showing that breastfeeding seems to counteract the effect of ambient influences on the growing lungs. Although the impact of breastfeeding on given childhood pulmonary disorders seems to present enduring controversies, a recent birth cohort of healthy infants found a protective causal role of exclusive breastfeeding in the risk of asthma in later childhood, which was mediated through its interplay on the gut microbiome [21]. In contrast, longitudinal studies from infancy to late adolescence have failed to prove the favorable effect of breastfeeding on lung function and respiratory morbidity in later life [22,23]. Thus, the impact of prolonged exclusive breastfeeding (PEB) on the lung function in childhood remains largely unknown.

The objective of this study was to explore the possible associations between breastfeeding and the lung function of children who were exclusively breastfed for an extensive period of time, in particular for more than 12 months. We hypothesized that PEB may have a propitious impact on the usual spirometric indices, independent of other confounding factors.

## 2. Materials and Methods

### 2.1. Study Design and Population

This cross-sectional, observational study was nested within the initiative ‘Children’s Lungs: from infancy to adolescence’ of the Pediatric Respiratory Unit of Democritus University of Thrace, Greece, which included children up to 14 years of age. The research model applied to conduct the present study was quantitative research methodology following a positivist approach, in order to test our hypothesis that PEB may have a beneficial effect on the spirometric parameters, independent of other confounding determinants. The study population was recruited from a private breastfeeding clinic in Thessaloniki, Greece, and children from 4 to 14-years-old who had been exclusively breastfed for at least 12 months were included. The lower age limit was chosen based on the possibility to perform a technically acceptable and repeatable spirometry. Children with congenital heart disease, neuromuscular disorders, chromosomal aberrations, or a history of significant prematurity (born at <34 weeks of gestation) were excluded. The parents of eligible participants were initially contacted via telephone and, if they agreed to participate, a visit to the clinic was scheduled. On the day of the visit, following a detailed explanation of the study aims and protocol by the investigators (EP, MM, AP), a consent document was initially filled and signed by the parents. This document included the authorization to access and use all the relevant clinical data of the participants. The study protocol was reviewed and approved by the Ethics Committee of the Democritus University of Thrace (IRB no. 23927/2382/02.01.2017).

### 2.2. Protocol and Measurements

Participants were invited to attend the clinic at predetermined dates, once per week, between January and December 2019. After the demographics and medical history, including the duration of exclusive breastfeeding from the medical records of the clinic, were collected; weight, height, waist and hip circumference, and the triceps, biceps, subscapular, and suprailiac skinfolds were measured. The body mass index (BMI) was calculated as weight/height^2^ (kg/m^2^) and assessed according to the International Obesity Task Force (IOFT) normative data [24]; overweight was defined as a BMI within the 25 and 30 kg/m^2^ cutoffs, and obesity as a BMI higher than the 30 kg/m^2^ threshold. Waist-to-height (WtH) ratio was calculated as the ratio between waist circumference (cm) and body height (cm), and used to calculate central obesity according to the International Diabetes Federation (IDF) criteria [25]. Body fat percentage was calculated based on skinfolds measurements using the Durnin-Womersley method [26].

Participants performed spirometry using a Micro 5000 spirometer (Medisoft, Sorinnes, Belgium) according to the ERS/ATS guidelines [27]. Forced expiratory volume at 1 s (FEV1), the forced vital capacity (FVC), FEV1/FVC ratio, and the forced expiratory flow between 25 and 75% of FVC (FEF25-75%) values were assessed according to the Global Lung Initiative normative data [28]. After the baseline spirometry, all the children received a 400 mcg salbutamol inhaler and the measurement was repeated after 15 min. Bronchial reversibility was defined as a FEV1 change from the baseline (ΔFEV1) >12% [29].

### 2.3. Statistical Analyses

A priori sample size calculation was not performed due to the paucity of the relevant clinical data (i.e., lung function measurements in children who were breastfed for a prolonged period of time). Continuous variables were expressed as mean± standard deviation (SD) and range. Spearman’s correlation was used to assess the relationship between spirometric indices (FEV1, FVC, FEV1/FVC, FEF25-75%, ΔFEV1) and duration of exclusive breastfeeding. Multivariable linear regression was used to assess the above relationship after adjustment for confounding factors, such as sex, age, BMI (Kg/m^2^) and WtH ratio. The statistical significance for the present study was set to 0.05. Statistical analyses were performed using IBM SPSS version 27 (IBM Corp., Armonk, NY, USA).

## 3. Results

From a total of 2562 breastfed children in the records of the clinic, 1944 were within the target age range. Of them, 1431 were exclusively breastfed; 321 less than 6 months, 1040 between 6 and 12 months, and 70 more than 12 months. Of the latter, 5 did not fulfill the inclusion criteria (3 had congenital heart disease and 2 were born preterm), 11 were not possible to contact (change of telephone number), and 8 did not consent to participate. Thus, the study population consisted of 46 children (21 boys) aged 9.2 ± 2.4 years, of whom the demographic and somatometric characteristics are presented in Table 1. The duration of exclusive breastfeeding was 27.5 ± 12.5 months (range 12 to 60 months). According to BMI IOFT norms, 13% of them were overweight (none were obese), while 21.7% had central obesity (IDF criteria). None of the children had a diagnosis of wheeze/asthma, neither had they ever received relevant medication.

Spirometric data are presented in Table 2. All the children had spirometric indices within the normal predictive range; the average FEV1 was 104.7 ± 10.4% (range 87–132%) and the average FEF25-75 was 107.9 ± 13.3% (range 88–145%). None of the participants had ΔFEV1 >12% (maximum ΔFEV1 8%).

No correlation was noted between the duration of breastfeeding and FEV1, FVC, or FEV1/FVC (Figure 1). In contrast, the duration of breastfeeding was significantly correlated with the FEF25-75 values (Spearman’s rho 0.422, *p* = 0.003). Multivariable linear regression analysis confirmed the above relationship, independent of the effect of sex, age, BMI, and WtH ratio (Table 3). In Table 3, we displayed two different models due to the fact that the BMI and WtH ratio cannot be entered together as they are correlated.

## 4. Discussion

The present study is the first to explore the association of breastfeeding on the lung function of children who were exclusively breastfed for an extended period; i.e., from at least 12 months and up to 5 years. We found that FEF25-75% was significantly related with the duration of breastfeeding, and that this correlation remained strong and significant after adjustment for sex, age, overweight, and central obesity. In addition, the children of our study were at a lesser degree overweighed and had less body fat in comparison to the general pediatric population [30,31]. Therefore, further to its favorable impact on the metabolic profile, PEB seems to exert a propitious effect on the function of smaller airways throughout childhood.

### 4.1. Relevant Studies

The protective effects of breastfeeding duration and exclusiveness on children’s health and early-life lung infections are well-documented; however, the long-term impact on lung function is still a matter of debate. Earlier studies have measured lung function at 4 years old and beyond, and have underscored the beneficial effect of breastfeeding in relation to lung growth and development [13]. Dogaru et al. [17] demonstrated that the forced mid-expiratory flow (FEF50%) was greater in children, at school age, who breastfed for more than 4 months compared to those who were fed with formula. Similarly, van Meel et al. [18] showed that non-exclusive breastfeeding for 4 months was associated with a lower FVC compared with those exclusively breastfed for 4 months, and shorter breastfeeding duration was associated with a lower FEV1 and FVC in school-age children. A systematic review concluded that prolonged and exclusive breastfeeding is beneficial for the lung function up to 18 years of age, showing an improved lung function, with the most consistent finding related to increased FVC [13], while studies in adults appeared conflicting towards supporting this long-lasting relationship [32,33]. In addition, large cross-sectional trials in children and adolescents with asthma have shown that breastfeeding is associated with a higher FEV1 and reduced number of exacerbations [34], and that the FEV1 and FVC values in later life are correlated significantly with the duration of breastfeeding [35]. A good knowledge of the beneficial effects of PEB may lead to the adaptation of efficacious promoting strategies.

In contrast, longitudinal studies have failed to show a favorable effect of breastfeeding on lung function. In the Promotion of Breastfeeding Intervention Trial (PROBIT), a breastfeeding promotion intervention had no detectable effect on the spirometric indices of the 13,557 participants after 16 years of follow-up [22]. Similarly, in the Avon Longitudinal Study of Parents and Children (ALSPAC), breastfeeding had no effect on the spirometric parameters, including bronchial hyper-responsiveness to methacholine, of the 13,978 participants at the age of 8 years [23]. However, as few as 3.6% of the enrolled children breastfed exclusively for more than 6 months in the PROBIT study [22], and only 8.5% for more than 4 months in the ALSPAC trial [23]. In the Melbourne Atopy Cohort Study (MACS) [36], 620 infants with a family history of atopy were recruited, and their lung function was assessed at 12 and 18 years of age. The duration of exclusive breastfeeding was in median 14 weeks, while only 2.6% of the participants were exclusively breastfed for more than 6.5 months. Interestingly, the duration of exclusive breastfeeding was marginally (*p* < 0.1) associated with FEF25-75% at 12 and 18 years of age; however, not with the FEV1 or FEV1/FVC ratio [36].

However, the previous published literature that has not found any positive associations regarding breastfeeding duration and lung function is sparse. It should be noted that in the aforementioned studies [22,23,36] the effect of PEB was most probably underestimated. First, the participants who were exclusively breastfed for more than 6 months represented only a small fraction of the study populations [22,23,36]. Second, exclusively breastfed children have been assessed together with their partially breastfed counterparts [22,23]; no subgroup analysis focusing on PEB has been performed; and thus, its potential effect may be masked. Third, it appears that the effect of breastfeeding cannot be readily detected by changes in FEV1 and FVC [22,23,36]. In contrast, the FEF25-75%, which represents a more sensitive marker of small airways dysfunction in the youths [37,38], seems to perform better in detecting subtle changes of lung function [36]. Taken together, the above data justify our decision to conduct a study focusing solely on PEB, even at the cost of a small sample size. In addition, the results of MACS [36] are in line with the herein reported correlation between the duration of exclusive breastfeeding and FEF25-75%; thus, lending further support to our approach.

### 4.2. Pathways through which Breastfeeding Effect Lung Function

The pathways by which breastfeeding may exert its beneficial effect on lung function remain largely unknown [13]. The effect might be mediated by reducing the number and complications of respiratory infections in infancy [13,14,15,16,39], and/or by protecting against allergic lung inflammation in early childhood [13,16,18,40]. The AAP and the WHO recommend that infants be ideally breastfed beyond the first year of life to attain the optimal short- and long-term outcomes, especially with regards to immune system maturation [6,41]. Complex underlying interactions between breastfeeding and a genetic predisposition towards asthma and/or atopy may also exist [13,39,42], while its role in the establishment and maintenance of the ‘intestinal microbiota-lung axis’ has recently gained much attention [10,43,44]. A recent large study of 2021 mother–child dyads, investigating the breastfeeding duration influence on childhood asthma development, showed a duration-dependent protective correlation of exclusive breastfeeding against ever asthma [45]. Finally, the positive effect of breastfeeding on lung function may be explained by its general beneficial effect on children’s health, including growth and development. Indeed, breastfeeding has been inversely associated with the risk of metabolic syndrome, overweight, and obesity in childhood [46,47,48]; and the investigators in MACS [36] showed that the association between the duration of exclusive breastfeeding and FEF25-75% was mediated by the somatometric characteristics of the participants, concerning better height and less BMI. In our children, PEB was also associated with a lower BMI, better WtH ratio, and less % fat, compared with the general national pediatric population [30,31]; however, our analyses showed that the effect of PBE on FEF25-75% was independent of these factors. Particularly, cross-sectional epidemiological studies estimated a prevalence of overweight and obesity among preadolescents in North-Eastern Greece and children aged 10 to 16 years old from Western Greece, equal to 48.5% and 28.1%, respectively, while central obesity, using IDF criteria, in Western Greece was 32.8% [30,31]. Although overweight, obesity, and central obesity seems to remain high among pediatric populations in Greece, among others, the advantageous effect of exclusive prolonged breastfeeding on the metabolic profile should be emphasized more through breastfeeding campaigns.

### 4.3. Study Limitations

Our study has several limitations. First, it is a single-center observational study, based on a relatively small sample size. Although as discussed above, we chose to solely focus on PEB at the cost of a restricted population sample, our results cannot be generalizable. On the other hand, the true effect of exclusive breastfeeding could only be assessed by a randomized clinical trial; yet, it would be unfeasible to randomize healthy infants to receive breastmilk or formula. Second, we lack control groups; e.g., of exclusively formula-fed infants or participants who were breastfed for less than 1 year. Again, our intention was to focus on PEB; the inclusion of control groups was deemed logistically unfeasible and beyond the aims of this nested study. Finally, we did not collect and, thus, we did not adjust for other possible confounders, such as parental socioeconomical status and tobacco smoke exposure.

## 5. Conclusions

In conclusion, the results of the present study show that PEB has a favorable effect on FEF25-75% throughout childhood, independent of known confounding factors, such as body composition, overweight, and obesity. FEF25-75% represents a sensitive parameter for assessing peripheral airway function among pediatric populations. Thus, our findings extend the existing knowledge on the propitious effects of exclusive breastfeeding, suggesting that it may also influence the function of smaller airways in the long-term. Taking into consideration our current findings, breastfeeding for more than 6 months should be regarded as a first step towards a healthy life and must be promoted, protected, and supported. Additional, well-designed studies are required to explore the mechanisms by which PEB may affect lung function in childhood and beyond.

## Figures and Tables

**Figure 1 children-09-01708-f001:**
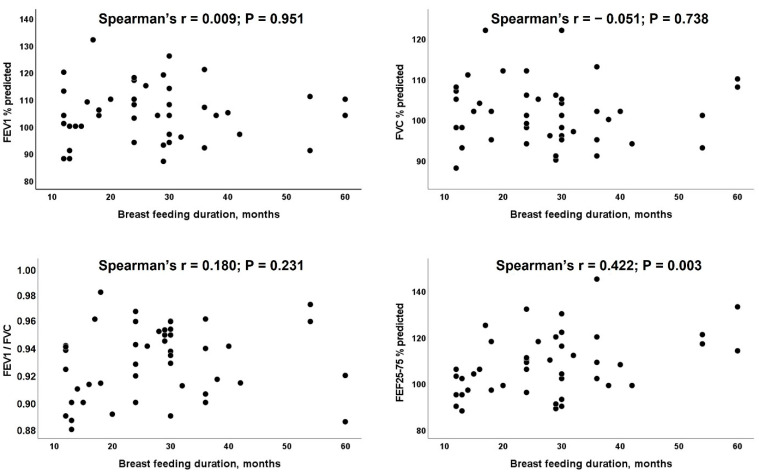
Correlation between duration of breast feeding and spirometric indices (N = 46).

**Table 1 children-09-01708-t001:** Demographic and somatometric characteristics.

N	46 *
Male/Female	21/25
Age, years	9.2 ± 2.4 (range: 4–14)
Weight, kg	30.9 ± 10.6 (range: 15.6–63.2)
Height, cm	133.2 ± 16 (range: 100–175)
BMI, kg/m^2^	16.9 ± 2.2 (range:14–23.4)
BMI percentile, % **	58.7 ± 17.8
BMI category, n (%) **	
Overweight (BMI 25–30 kg/m^2^)	6 (13%)
Obesity (BMI >30 kg/m^2^)	0 (0)
Waist circumference, cm	64.3 ± 9 (range: 49–81)
Waist-to-height ratio	0.48 ± 0.03 (range: 0.42–0.56)
Central obesity, n (%) ***	10 (21.7%)
Skin folds, mm	
Biceps	6.2 ± 2.6 (range: 3–13.3)
Triceps	10.2 ± 3.5 (range: 3.5–22.1)
Subscapular	5.9 ± 2 (range: 3.7–14.3)
Suprailiac	8.1 ± 4 (range: 3.3–17.7)
Body fat, % ****	19.7 ± 4.7 (range: 10.8–27.8)
Breastfeeding duration, months	27.5 ± 12.5 (range: 12–60)

Data are mean ± SD (range); * Data collected from January to December 2019; ** International Obesity Task Force (IOFT) normative data and classification; *** International Diabetes Federation (IDF) classification; **** Durnin-Womersley Caliper method.

**Table 2 children-09-01708-t002:** Spirometric characteristics (N = 46) *.

FEV1, % predicted	104.7 ± 10.4 (range: 87–132)
FVC, % predicted	101.3 ± 7.7 (range: 88–122)
FEV1/FVC	0.93 ± 0.03 (range: 0.88–0.98)
FEF25-75%, % predicted	107.9 ± 13.3 (range: 88–145)
ΔFEV1, %	3.6 ± 1.7 (range: 1–8)

Data are mean ± SD (range). Predicted values calculated according to Global Lung Initiative (GLI) norms.; * Data collected from January to December 2019.

**Table 3 children-09-01708-t003:** Dependencies of FEF25-75 % predicted (log) (N = 46) *.

	Model 1	Model 2
Male sex	0.033 (0.823)	0.091 (0.529)
Age	0.084 (0.564)	0.081 (0.578)
BMI percentile	0.123 (0.407)	-
Waist-to-height ratio	-	0.110 (0.450)
Breastfeeding duration	0.477 (0.002)	0.478 (0.002)

Multivariable linear regression analyses with log FEF25-75% predicted as dependent variable. Data are regression coefficients with *p* values in parentheses.; * Data collected from January to December 2019.

## Data Availability

The data supporting the findings of this study are available within the article and its tables/figures; raw data are available upon reasonable request from the corresponding author.

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
