# Peer review of "Association between History of Prolonged Exclusive Breast-Feeding and the Lung Function Indices in Childhood"

_children, 2022, doi:10.3390/children9111708_

Round 1

Reviewer 1 Report

Dear authors, first of all, I congratulate you for addressing this critical topic. Second, regarding the article, it must be improved before publishing it.

Title: Ok.

Summary:

MeSH: It has no keywords.

Introduction:

The problematization of the subject to be studied must be improved; figures are lacking in this regard, and the support of the main actors in the issue, such as WHO. Likewise, the objective must be clearly stated, which must be the same as in the summary.

Methodology: The research paradigm is not mentioned, although it is intuited that it is positivist. It would give more support to the research to use a methodological reference for the type of study due to the different ways of approaching a quantitative design.

Study type: it must be declared who authorized access to the data for the study invitation.

Participants: the total study population must be declared, the type of sample design used if there were refusals to participate, and identifying reliability and statistical power.

Data collection must be declared at what time the informed consent was applied, by whom, and how it was applied to each family.

Data analysis: all the study variables must be declared clearly, thus complementing the type of analysis.

Results:

The title of the tables and figures must be completed with data such as the total population number, date, and place of data collection. As well as place the authorship of elaboration of these.

The variables must be conceptualized in methodology to understand their presentation in results.

Discussion:

Due to the large amount of data presented in the results, it is advisable to separate the discussion with subtitles to facilitate reading.

Limitations are declared.

It is necessary to improve the comparison of all the results with other studies.

Conclusions: It does not clearly show the contribution to the discipline.

References: must be updated as more than 50% are more than five years old, in addition to three concerns from 2017 that will soon also be more than five years old. Likewise, the main actors in the issue, such as WHO, should be included.

Author Response

Reviewer 1

 We wish to thank the Reviewer for the time spent to review our paper and for the valuable comments that helped us to improve our manuscript. A point-by- point answer to all comments follows:

Comment #1: Summary, MeSH: It has no keywords.

Authors’ response: Keywords have been included following the abstract section; please refer to 'Keywords’ (breastfeeding; spirometry; children), L36.

Comment #2: Introduction: The problematization of the subject to be studied must be improved; figures are lacking in this regard, and the support of the main actors in the issue, such as WHO.

Authors’ response: Thank you for your comment. To address this issue, we added information on the main actors in the field, such as American Academy of Pediatrics (AAP), World Health Organization (WHO), and United Nations Children’s Fund (UNICEF). Please refer to L44-65.

Comment #3: Likewise, the objective must be clearly stated, which must be the same as in the summary.

Authors’ response: We have rephrased our objective in introduction section, which is the same with that in the abstract section (L82-84).

Comment #4: Methodology: The research paradigm is not mentioned, although it is intuited that it is positivist. It would give more support to the research to use a methodological reference for the type of study due to the different ways of approaching a quantitative design.

Authors’ response: Thank you for your comment, we made the methodological clarifications, as requested (L89-94).

Comment #5: Study type: it must be declared who authorized access to the data for the study invitation.

Authors’ response: The study was approved by the Ethics Committee of the Democritus University of Thrace (IRB No. 23927/2382/02.01.2017) and written informed consent was obtained from the parents prior to enrollment. The consent document included the authorization to access and use all relevant clinical data (L100-107).

Comment #6: Participants: the total study population must be declared, the type of sample design used if there were refusals to participate, and identifying reliability and statistical power.

Authors’ response: Thank you for your comment. Details about study population, refusals, etc, were included in the first paragraph of the ‘Results’ (L139-144).

A priori sample size calculation was not performed due to the paucity of relevant clinical data (i.e., lung function measurements in children who were breastfed for a prolonged period of time). This is now stated in ‘Statistics’ (L130-132).

Comment #7: Data collection must be declared at what time the informed consent was applied, by whom, and how it was applied to each family.

Authors’ response: Thank you for your comment, we performed the requested clarifications in the ‘Methods’ (L100-105).

Comment #8: Data analysis: all the study variables must be declared clearly, thus complementing the type of analysis.

Authors’ response: Thank you for your comment. We performed the requested changes in ‘Statistics’ (L132-138).

Comment #9: Results: The title of the tables and figures must be completed with data such as the total population number, date, and place of data collection. As well as place the authorship of elaboration of these.

Authors’ response: Thank you for this comment. The number of participants (N) was added in table and figure titles. The rest of the requested information, when possible, was added in the footnotes. We would like to note, however, that the interested reader can find all relevant information in the ‘Methods’.

Comment #10: The variables must be conceptualized in methodology to understand their presentation in results.

Authors’ response: All variables have been defined in detail in the ‘Methods’ (L123-128).

Comment #11: Discussion: Due to the large amount of data presented in the results, it is advisable to separate the discussion with subtitles to facilitate reading.

Authors’ response: Thank you for this comment. We separated the discussion with appropriate subtitles.

Comment #12: It is necessary to improve the comparison of all the results with other studies.

Authors’ response: Thank you for your comment. We have made several additions to improve the comparison with other relevant studies (L177-225 & L245-254).

Comment #13: Conclusions: It does not clearly show the contribution to the discipline. Authors’ response: Thank you for your comment. We have added a sentence showing the major contribution of FEF25-75% in lung function in children. Also, we have added that in light of our results, breastfeeding for more than 6 months should be promoted (L269-278).

Comment #14: References: must be updated as more than 50% are more than five years old, in addition to three concerns from 2017 that will soon also be more than five years old. Likewise, the main actors in the issue, such as WHO, should be included.

Authors’ response Thank you for your comment, we have added the main actors in the field, such as American Academy of Pediatrics (AAP), World Health Organization (WHO), and United Nations Children’s Fund (UNICEF) in introduction section. Regarding references, we didn't delete any of the already existence references because their all pioneering to the field, regardless of their published date, although to address this issue, and to increase the readability giving a deeper understanding of the research content, we have added the following references:

  1. Meek, J.Y.; Noble, L. Technical Report: Breastfeeding and the Use of Human Milk. Pediatrics 2022, 150, e2022057989. doi: 10.1542/peds.2022-057989.
  2. World Health Organization. Ten steps to successful breastfeeding. https://www.who.int/teams/nutrition-and-food-safety/food-and-nutrition-actions-in-health-systems/ten-steps-to-successful-breastfeeding
  3. Iliodromiti, Z.; Zografaki, I.; Papamichail, D.; Stavrou, T.; Gaki, E.; Ekizoglou, C.; Nteka, E.; Mavrika, P.; Zidropoulos, S.; Panagiotopoulos, T.; Antoniadou, I. Increase of breast-feeding in the past decade in Greece, but still low uptake: cross-sectional studies in 2007 and 2017. Public Health Nutr. 2020, 23, 961-970. doi:10.1017/S1368980019003719.
  4. Tigka, M.; Metallinou, D.; Nanou, C.; Iliodromiti, Z.; Lykeridou, K. Frequency and Determinants of Breastfeeding in Greece: A Prospective Cohort Study during the COVID-19 Pandemic. Children 2022, 9, 43. doi: 10.3390/children9010043.
  5. Moshammer, H.; Hutter, H.P. Breast-Feeding Protects Children from Adverse Effects of Environmental Tobacco Smoke. Int. J. Environ. Res. Public Health. 2019, 16,304. doi: 10.3390/ijerph16030304.
  6. Rosas-Salazar, C.; Shilts, M.H.; Tang, Z.Z.; Hong, Q.; Turi, K.N.; Snyder, B.M.; Wiggins, D.A.; Lynch, C.E.; Gebretsadik, T.; Peebles, R.S.Jr.; Anderson, L.J.; Das, S.R.; Hartert, T.V. Exclusive breast-feeding, the early-life microbiome and immune response, and common childhood respiratory illnesses. J. Allergy Clin. Immunol. 2022, 150, 612-621. doi: 10.1016/j.jaci.2022.02.023.
  7. Wilson, K.; Gebretsadik, T.; Adgent, M.A.; Loftus, C.; Karr, C.; Moore, P.E.; Sathyanarayana, S.; Byington, N.; Barrett, E.; Bush, N.; Nguyen, R.; Hartman, T.J.; LeWinn, K.Z.; Calvert, A.; Mason, W.A.; Carroll, K.N. The association between duration of breastfeeding and childhood asthma outcomes. Ann. Allergy Asthma Immunol. 2022, 129, 205-211. doi: 10.1016/j.anai.2022.04.034

Reviewer 2 Report

Could the authors be so kind and rewrite the methods of their study in a more clear way.

Could the authors explain how did they decided to choose the lowest and highest age for their study group 

Could the authors also comment on the regular practice of breastfeeding in their country

Author Response

Reviewer 2

 We wish to thank the Reviewer for the time spent to review our paper and for the valuable comments that helped us to improve our manuscript. A point-by-point answer to each comment follows:

Comment #1: Could the authors be so kind and rewrite the methods of their study in a more clear way. Authors’ response: Thank you for your comment. We have made significant changes and additions in Materials and Methods section, hopping that we have been achieved a clearer writing.

Comment #2: Could the authors explain how they decided to choose the lowest and highest age for their study group. 

Authors’ response: Thank you for your comment. This study was nested within the initiative ‘Children’s Lungs: from infancy to adolescence’ of the Pediatric Respiratory Unit of Democritus University of Thrace, Greece, which was originally designed to include children up to 14 years of age. The lower age limit was chosen based on the possibility to perform a technically acceptable and repeatable spirometry. We added the requested information in the manuscript (study design and population section).

Comment #3: Could the authors also comment on the regular practice of breastfeeding in their country.

Authors’ response: Thank you for your comment. We added what is considered the 'norm' in Greece in introduction section, using the necessary references (Tigka, M.; et al. and Iliodromiti, Z.; et al.)

Reviewer 3 Report

The topic is very interesting because it deals with evaluating the effects at a distance, a very demanding goal and also for this reason little attended.

The respiratory outcome is of particular relevance.

It is very dry, essential in the explanation.

Some more detailed reference to the recognized effects of exclusive feeding with breast milk and respiratory outcome seems useful and necessary to me.

The evaluation of the incidence of overweight and obesity was also considered in the structuring of the study. It should be emphasized more. It is true that the filter of confounding factors has not been applied, but it seems to me to be a non-marginal element.

In the introduction there are some considerations identical to those of the discussion for which one must choose where to leave them.

The choice of the target age and the real one is not well understood. It should be better detailed.

I would add the fact that, in light of the results, breastfeeding for more than 6 months should be promoted.

It should also be emphasized that the previous literature that has not found positive associations is however sparse, since it is a few studies.

Author Response

Reviewer 3

We wish to thank the Reviewer for the time spent to review our paper and for the valuable comments that helped us to improve our manuscript. A point-by-point answer to each comment follows: 

Comment #1. Some more detailed reference to the recognized effects of exclusive feeding with breast milk and respiratory outcome seems useful and necessary to me.

Authors’ response: Thank you for your comment. We made additions in the introduction section, citing of course related work (Rosas-Salazar, C.;et al, and Moshammer, H.; Hutter, H.P.)

Comment #2. The evaluation of the incidence of overweight and obesity was also considered in the structuring of the study. It should be emphasized more. 

Authors’ response: Thank you for your comment, in order to address this issue, we made changes, providing more information about national prevalence of overweight, obesity and central obesity in our country, in discussion section.

Comment #3. It is true that the filter of confounding factors has not been applied, but it seems to me to be a non-marginal element.

Authors’ response: Known confounders, such as age, BMI, and central obesity were tested in the multivariable models of Table 3. We agree with the Reviewer that many other confounding factors could exist; however, those data were not originally collected.

Comment #4. In the introduction there are some considerations identical to those of the discussion for which one must choose where to leave them.

Authors’ response: Thank you for your comment, we deleted from introduction those considerations which were similar to those in discussion.

Comment #5. The choice of the target age and the real one is not well understood. It should be better detailed.

Authors’ response: Thank you for your comment. This study was nested within the initiative ‘Children’s Lungs: from infancy to adolescence’ of the Pediatric Respiratory Unit of Democritus University of Thrace, Greece, which was originally designed to include children up to 14 years of age. The lower age limit was chosen based on the possibility to perform a technically acceptable and repeatable spirometry. We added the requested information in study design and population section. We added the requested explanation in the section of study design and population.

Comment #6. I would add the fact that, in light of the results, breastfeeding for more than 6 months should be promoted.

Authors’ response: Thank you for your comment. We added this fact to the conclusion section, as well as to the introduction; in which we have added the recommendations of American Academy of Pediatrics (AAP) and World Health Organization (WHO) regarding desirable breastfeeding duration.

Comment #7. It should also be emphasized that the previous literature that has not found positive associations is however sparse, since it is a few studies.

Authors’ response: Thank you for your comment. We have added a similar sentence in the appropriate place in discussion section.

Round 2

Reviewer 2 Report

The authors have added the requested informations.  There are still some minor spelling issue that have to be corrected

Author Response

We would like to thank the reviewer for this valuable comment, which helps in improving the quality of the manuscript.

We have made additional spelling corrections in the re-submitted manuscript